# A Replication Study Identified Seven SNPs Associated with Quantitative Traits of Type 2 Diabetes among Chinese Population in A Cross-Sectional Study

**DOI:** 10.3390/ijerph17072439

**Published:** 2020-04-03

**Authors:** Fan Yuan, Hui Li, Chao Song, Hongyun Fang, Rui Wang, Yan Zhang, Weiyan Gong, Ailing Liu

**Affiliations:** National Institute for Nutrition and Health, Chinese Center for Disease Control and Prevention, Beijing 100050, China; yuanfan@ninh.chinacdc.cn (F.Y.); abclihui@163.com (H.L.); songchao@ninh.chinacdc.cn (C.S.); fanghy@ninh.chinacdc.cn (H.F.); wangrui@ninh.chinacdc.cn (R.W.); zhangyan@ninh.chinacdc.cn (Y.Z.); gongwy@ninh.chinacdc.cn (W.G.)

**Keywords:** type 2 diabetes, insulin resistance, impaired insulin release, SNP, HOMA-IR, HOMA-β

## Abstract

Genome-wide association studies (GWAS) have identified common variants for quantitative traits (insulin resistance and impaired insulin release) of type 2 diabetes (T2D) across different ethnics including China, but results were inconsistent. The study included 1654 subjects who were selected from the 2010–2012 China National Nutrition and Health Surveillance (CNNHS). Insulin resistance and impaired insulin release were assessed by homeostasis model assessment (HOMA). The study included 64 diabetes-related single nucleotide polymorphisms (SNPs), which were done using Mass ARRAY. A logistic regression model was employed to explore the associations of SNPs with insulin resistance and impaired insulin release by correcting for the confounders. The 5q11.2-rs4432842, RASGRP1-rs7403531, and SEC16B-rs574367 increased the risk of insulin resistance with OR = 1.23 (95% CI: 1.04–1.45, OR = 1.35 (95% CI: 1.13–1.62), OR = 1.34 (95% CI: 1.07–1.67), respectively, while MAEA-rs6815464 decreased the risk of insulin resistance (OR = 0.84, 95% CI: 0.71–1.00). CENTD2-rs1552224, TSPAN8-rs7961581 and ANK1-rs516946 was associated with increased risk of impaired insulin release with OR = 1.47 (95% CI: 1.09–1.99), OR = 1.25 (95% CI: 1.03–1.51), OR = 1.39 (95% CI: 1.07–1.81), respectively. Our findings would provide insight into the pathogenesis of individual SNPs and T2D.

## 1. Introduction

Type 2 diabetes (T2D) is a severe chronic non-communicable disease concerning public health in China [1]. It may cause multiple organ complications such as diabetic retinopathy, diabetic nephropathy, and peripheral neuropathy, which has become a burden not only for individual physiology and psychology but also for allocation of health resources [2]. The pathogenesis of T2D is complicated, although lifestyle is a great contributor to the T2D epidemic, genetic determinants can also affect the T2D susceptibility [3]. Individuals carrying risk alleles predisposed to T2D are mediated by insulin resistance or abnormal insulin secretion [4]. Insulin resistance [5] is one critical process to the development of T2D, which means the function of pancreatic β-cells is commonly normal and insulin secretion is sufficient, but the insulin receptors are insensitive to insulin levels and the ability to stimulate glucose utilization is decreasing, or impairments in insulin signal transduction, resulting in a compensatory hyperinsulinemia [6]. Besides, insulin resistance is ordinarily related to obesity or other metabolic syndromes [7].

Insulin release decrease is another mechanism of T2D when pancreatic β-cells are impaired, and individuals are unable to utilize glucose which will cause high postprandial blood glucose. A majority of variants have been identified to associate with β-cell dysfunction [8], but evidence suggests that Asians might be more susceptible to insulin resistance than Europeans [9]. Genome-wide association studies (GWAS) for T2D have been widely carried out in different racial types, with the exception of T2D-associated genes, obesity-associated and birth weight-associated genes have also been found to relate to T2D, but results were inconsistent. For example, loci rs13266634 in gene SLC30A8 was associated with T2D in Russian and the Danish populations [10,11], while the link disappeared in African populations [12]. A common obesity-associated gene, FTO, has been confirmed to have an association with T2D in different countries, and several gene loci were associated with insulin resistance such as rs9939609 and rs1558902 [13,14]. Birth weight-associated loci rs6931514 in gene CDKAL1 was associated with reduced insulin secretion in Europeans [15], which suggested that the genetic variation was affected differently among races and geographic areas [2]. Insulin resistance and insulin release were assessed by homeostasis model assessment (HOMA) in multiple studies based on fasting plasma glucose (FPG) and fasting serum insulin (FSIN) [4]. Thus, our aim was to replicate the relationship that has been confirmed in other ethnic groups, further, to explore novel loci that have never been clearly associated with T2D-related quantitative traits in the Chinese population, providing the pathogenesis of T2D.

## 2. Materials and Methods

### 2.1. Study Subjects

The data were from the 2010–2012 China National Nutrition and Health Surveillance (CNNHS). The CNNHS was a nationally representative cross-sectional surveillance conducted by the National Institute for Nutrition and Health, Chinese Center for Disease Control and Prevention (NINH, China CDC), which covered 31 provinces, autonomous regions, and municipalities (except for Taiwan, Hong Kong, and Macao). Cluster random sampling with multi-stage stratification proportional to population size sampling design was used in recruiting subjects [16]. For the present study, subjects born in 1960, 1961, and 1963 were included, and those with a low call rate of variants (<50%), unqualified blood samples, failed DNA extraction, abnormal gene detection results, incomplete basic information, and those with T2D who took control measures were excluded. Finally, a total of 1654 subjects were included in the present study. The protocol was approved by the Ethics Committee of NINH, China CDC (No.2013-010). Written informed consent was obtained from each subject. 

### 2.2. Assessment and Definition of Insulin Resistance and Impaired Insulin Release

Fasting venous blood samples were taken from each subject in the morning. Fasting plasma glucose (FPG) was measured using the glucose oxidize enzymatic method within 3 hours of blood collection. Fasting serum insulin (FSIN) was measured by radioimmunoassay. Insulin resistance and impaired insulin release were assessed by homeostasis model assessment of insulin resistance (HOMA-IR), homeostasis model assessment of beta cell function (HOMA-β), respectively, which was calculated by the following equation:

HOMA-IR = FPG × FSIN/22.5, HOMA-β = FSIN × 20/(FPG − 3.5) (with serum insulin in mU/L and plasma glucose in mmol/L) [4].

Insulin resistance was defined by exceeding the top quartile of HOMA-IR index value and impaired insulin release was defined under bottom quartile of HOMA-β index value in non-diabetic individuals [17].

### 2.3. Genotyping and SNP Selection

The genotype of 81 diabetes-related single nucleotide polymorphisms (SNPs) were detected by Mass ARRAY (Agena, San Diego, CA, USA). SNPs’ exclusion criteria: (1) call rates <80%; (2) deviating from Hardy–Weinberg equilibrium *p* < 0.001 (Appendix A); (3) a minor allele frequency (MAF) of each SNP <5%. Finally, a total of 64 variants were involved in the present study, of which 22 SNPs had previously confirmed association with insulin resistance (Appendix A), 27 loci were associated with insulin release (Appendix A). The association of the left 15 loci with T2D had been indicated, however, whether these SNPs related to insulin release or insulin resistance were still unclear.

### 2.4. Demographic Characteristic, Physical Activity and Dietary Measurement

Questionnaires were used to obtain information. Self-reported educational levels were categorized into three levels: primary school or lower, junior middle school, senior high school or above. According to the per capita annual income of urban and rural households in 2011, the family economic level was divided into three levels: low (<20,000 RMB), middle (20,000–40,000 RMB) and high (>40,000 RMB). Smoking was coded as “yes” if smoking during the past 30 days. Drinking alcohol was coded as “yes” if drinking any type of alcohol during the past 12 months. Physical exercise was coded as “yes” if attending leisure time physical exercise during the past 3 months. The sedentary time was defined as time spent sitting or lying in leisure time. 

Height, weight, and waist circumference was measured by trained investigators under standard operation procedure. Body mass index (BMI) was calculated as weight in kilograms divided by height in meters squared (kg/m^2^).

The dietary intake was collected by food frequency questionnaire (FFQ) and three consecutive (two weekdays and one weekend day) 24-h recalls which were described elsewhere [16]. The whole cereal and beans intake level was divided into low (less than 50 g per day), medium (more than or equal to 50 g and less than or equal to 150 g per day), and high (more than 150 g per day). Livestock and poultry intake level was divided into low (less than 40 g per day), medium (more than or equal to 40 g and less than or equal to 75 g per day), and high (more than 75 g per day) based on the recommendations of Dietary Guideline for Chinese Residents (2016) [18]. 

### 2.5. Statistical Analysis

Data analyses were conducted using statistical analysis system (SAS) 9.4 software (SAS Institute, Cary, NC, USA). A Hardy–Weinberg equilibrium test was performed for each SNP of non-diabetic individuals by chi-square test [19]. Continuous variables were expressed as mean ± SD, and categorical variables were presented as frequency and percentage. T-tests and chi-square tests were performed for comparisons of continuous and categorical variables between males and females, respectively. A logistic regression model was utilized to validate the association of individual SNPs with T2D-related quantitative traits corrected for confounders, including gender, economical status, education levels, BMI, waist circumference, sedentary time, exercise, whole cereal and beans intake levels, livestock and poultry intake levels, drinking, and smoking. Two-tailed *p* < 0.05 regarded as statistical significance.

## 3. Results

### 3.1. Basic Characteristics of Study Subjects

Basic characteristics are shown in Table 1. The age of 1654 subjects (male 39.1%, female 60.9%) was 49.9 ± 1.5 years. The rate of impaired insulin release in males was 32.6%, which was higher than that in females (*p* < 0.05). There were no significant differences between male and female in the rate of insulin resistance. There were significant differences between males and females in education level, smoking, drinking, whole cereal and beans intake levels, livestock and poultry intake levels, BMI, waist circumference, and exercise.

### 3.2. Association between Individual SNP and Insulin Resistance 

As shown in Table 2 and Figure 1, three SNPs were associated with increased insulin resistance risk with adjusting for covariates, including 5q11.2-rs4432842 (OR = 1.23, 95% CI: 1.04–1.45), RASGRP1-rs7403531 (OR = 1.35, 95% CI: 1.13–1.62), SEC16B-rs574367 (OR = 1.34, 95% CI: 1.07–1.67). Rs11030104 in gene BDNF (OR = 1.18, 95% CI: 1.01–1.39) with insulin resistance was found, which disappeared after adjusting for covariates (*p* = 0.184). 

MAEA-rs6815464 was associated with decreased insulin resistance with correction for covariates, the OR was 0.84 (95% CI: 0.71–1.00). CENTD2-rs1552224 had an association with decreased insulin resistance (OR = 0.75, 95% CI: 0.58–0.98), while the link disappeared after controlling for covariates (*p* = 0.063).

### 3.3. Association between Individual SNP and Impaired Insulin Release 

Association of CENTD2-rs1552224 (OR = 1.47, 95% CI: 1.09–1.99), TSPAN8-rs7961581 (OR = 1.25, 95% CI: 1.03–1.51) and rs516946 (OR = 1.39, 95% CI: 1.07–1.81) with increased impaired insulin release were significant without or with adjustment for covariates. (See Table 2 and Figure 2)

## 4. Discussion

Among the 64 susceptible loci we examined, RASGRP1-rs7403531, ANK1-rs516946, and SEC16B-rs574367 had associations with insulin resistance. HHEX-rs5015480, CENTD2-rs1552224, and TSPAN8-rs7961581 were associated with impaired insulin release in the current Chinese population.

Our study has replicated the association of rs7403531 with insulin resistance, which was consistent with a previous study which reported that rs7403531 had been associated with T2D and higher insulin in Chinese Hans [20]. However, Sakai et al. [21] found no significant association of rs7403531 with T2D. Insulin sensitivity is one of the predictors of diabetes [22], therefore, these findings indicated that carriers of rs7403531 may be predisposed to T2D by decreasing insulin sensitivity. Additionally, rs7403531 had the same OR for insulin resistance after correcting for BMI, suggesting its association with insulin resistance might not be mediated through obesity.

Most studies were conducted to explore the association of rs574367 with BMI, but seldom T2D. The G allele of SEC16B-rs574367 increased the risk for obesity in different ethnic populations [23]. SEC16B-rs574367 could affect the synthesis and transcription of lipase, thus inhibiting the decomposition of fat, leading to the occurrence of obesity [24]. Several obesity-associated loci including rs574367 constructed a BMI gene risk score (BMI-GRS), which had an enhanced effect on log (HOMA2-IR) in the age- and sex-adjusted model among Chinese adults [25]. Our study found rs574367 had the same risk for increased HOMA-IR. Further study should be conducted to confirm the relationship.

MAEA-rs6815464 (macrophage erythroblast attacher) was reported as a T2D risk variant in Asians [26,27]. In the Chinese population, no significant association was observed with insulin resistance or insulin release, measured by fasting-based homeostasis model assessment [28]. A genome-wide association study identified MAEA gene altered T2D risk through insulin secretion [29]. However, in our study, rs6815464 had an association with decreased insulin resistance after adjustment for confounding factors. Therefore, further functional characterization study is required to elucidate its role in the pathogenesis of T2D.

A Mendelian randomization study has shown that lowered birthweight 5q11.2-rs4432842 could increase susceptibility to T2D [30]. However, it remains unclear what its role in the pathogenesis of T2D is. Our study found 5q11.2-rs4432842 was associated with increased risk of insulin resistance.

The association of rs516946 with T2D has been confirmed by previous studies [31,32]. ANK1-rs516946 was associated with impaired insulin release by decreasing insulinogenic and disposition indexes in a Danish cohort study [32]. The risk C allele of rs516946 with a larger waist circumference was found in a Han Chinese population [31]. After adjusting for waist circumference, the risk C allele of ANK1-rs516946 was associated with increased impaired insulin release in our study.

CENTD2-rs1552224 could increase the risk of T2D through reducing insulin secretion [11]. CENTD2-rs1552224 associated with decreased glucose-stimulated insulin release, which increased 30-min plasma glucose values by 2.0% and reduced insulin release 30 min after an oral glucose load by 4.2% [33]. CENTD2-rs1552224 increased the risk of impaired insulin release after correction for covariates in our study.

TSPAN8-rs7961581 increased the risk of T2D in a previous study in Han Chinese [34]. A study in Denmark confirmed that the C allele of TSPAN8-rs7961581 associated with decreased levels of corrected insulin response, of AUC-insulin/AUC-glucose ratio, and of the insulinogenic index [35]. Our study has replicated the association of TSPAN8-rs7961581 with T2D, TSPAN8-rs7961581 increased the risk of impaired insulin release by 25%.

The present study had some limitations. Firstly, the method of HOMA was not the golden standard to assess insulin resistance and impaired insulin release. The standard intravenous methods (euglycemic insulin clamp technique) and comprehensive indices should be used in the Chinese population to validate the relationship in future studies. Secondly, the participants in the present study were about 50 years old and too young to get enough cases of T2D. The small numbers may partly explain why no significant associations of some SNPs with the risk of quantitative traits of T2D were found. 

## 5. Conclusions

In summary, we have replicated the association of seven SNPs and quantitative traits of T2D among the Chinese population, of which 5q11.2-rs4432842, RASGRP1-rs7403531, and SEC16B-rs574367 were associated with increased insulin resistance, MAEA-rs6815464 was associated with decreased insulin resistance, and ANK1- rs516946, CENTD2-rs1552224, and TSPAN8-rs7961581 were associated with increased impaired insulin release. Our results would provide insight into the pathogenesis of individual SNP and T2D.

## Figures and Tables

**Figure 1 ijerph-17-02439-f001:**
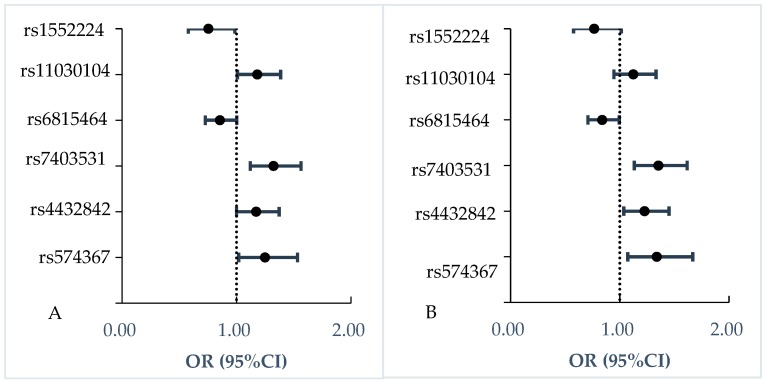
Association of six SNPs with insulin resistance. (**A**) Model 1 was logistic regression model to analyze the association of individual SNP with quantitative traits without adjusting for any covariates. (**B**) Model 2 was logistic regression model to analyze the association of individual SNP with quantitative traits adjusting for gender, economical status, education levels, BMI, waist circumference, sedentary time, exercise, whole cereal and beans intake levels, livestock and poultry intake levels, drinking, and smoking.

**Figure 2 ijerph-17-02439-f002:**
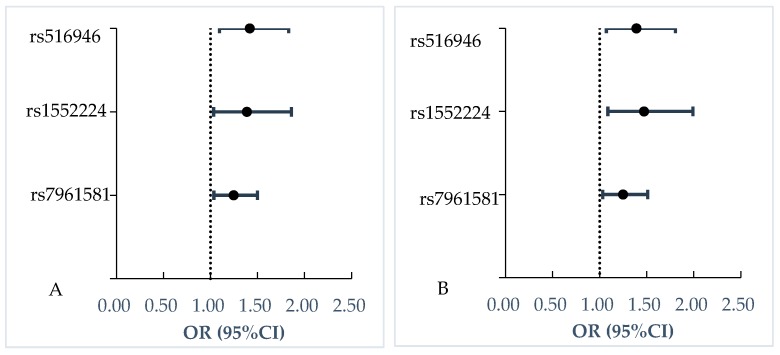
Association of three SNPs with impaired insulin release. (**A**) Model 1 was logistic regression model to analyze the association of individual SNP with quantitative traits without adjusting for any covariates. (**B**) Model 2 was logistic regression model to analyze the association of individual SNP with quantitative traits adjusting for gender, economical status, education levels, BMI, waist circumference, sedentary time, exercise, whole cereal and beans intake levels, livestock and poultry intake levels, drinking, and smoking.

**Table 1 ijerph-17-02439-t001:** Basic characteristics of 1654 subjects.

Characteristics	Total	Male	Female	*p* Value
Total	1654	647 (39.1%)	1007 (60.9%)	
Age (year)	49.9 ± 1.5	49.9±1.5	49.9 ± 1.5	0.9347
**Insulin resistance**				0.0950
No	1189 (71.9%)	480 (74.2%)	709 (70.4%)	
Yes	465 (28.1%)	167 (25.8%)	298 (29.6%)	
**Impaired insulin release**				0.0050
No	1179 (71.3%)	436 (67.4%)	743 (73.8%)	
Yes	475 (28.7%)	211 (32.6%)	264 (26.2%)	
**Education level**				<0.0001
Illiterate or primary school	563 (34%)	140 (21.6%)	423 (42.0%)	
Middle school	723 (43.7%)	327 (50.5%)	396 (39.3%)	
Senior school	368 (22.2%)	180 (27.8%)	188 (18.7%)	
**Economic status**				0.7453
Low	839 (52.7%)	324 (52.1%)	515 (53.1%)	
Middle	633 (39.8%)	254 (40.8%)	379 (39.1%)	
High	119 (7.5%)	44 (7.1%)	75 (7.7%)	
**Smoking**				<0.0001
No	1164 (70.5%)	204 (31.6%)	960 (95.4%)	
Yes	487 (29.5%)	441 (68.4%)	46 (4.6%)	
**Drinking**				<0.0001
Yes	1121 (67.9%)	250 (38.7%)	871 (86.6%)	
No	531 (32.1%)	396 (61.3%)	135 (13.4%)	
**Whole cereal and beans intake levels**				0.0171
Low	1080 (86.5%)	425 (88%)	655 (85.5%)	
Medium	133 (10.6%)	39 (8.1%)	94 (12.3%)	
High	36 (2.9%)	19 (3.9%)	17 (2.2%)	
**Livestock and poultry intake levels**				0.0003
Low	517 (41.4%)	184 (38.1%)	333 (43.5%)	
Medium	284 (22.7%)	93 (19.3%)	191 (24.9%)	
High	448 (35.9%)	206 (42.7%)	242 (31.6%)	
**BMI (kg/m^2^)**	24.3 ± 3.4	24.0 ± 3.4	24.5 ± 3.4	0.002
**Waist circumference (cm)**	82.2 ± 10.0	83.7 ± 10.4	81.3 ± 9.6	<0.0001
**Exercise**				0.0409
No	1500 (91.4%)	600 (93.2%)	900 (90.3%)	
Yes	141 (8.6%)	44 (6.8%)	97 (9.7%)	
**Sedentary time (h/d)**	2.7 ± 1.5	2.6 ± 1.4	2.7 ± 1.6	0.6235

Continuous variables were presented as mean ± SD, and categories variables were presented as N (%).

**Table 2 ijerph-17-02439-t002:** Association between each SNP and quantitative traits of T2D.

Identified Association to	Loci	Chr	Reported Gene	Risk/Other Allele	Insulin Resistance				Impaired Insulin Release			
					Model 1		Model 2		Model 1		Model 2	
					OR (95%CI)	*p* Value	OR (95%CI)	*p* Value	OR (95%CI)	*p* Value	OR (95%CI)	*p* Value
T2D	rs340874	1	PROX1	C/T	0.97(0.83,1.14)	0.718	0.96(0.81,1.13)	0.595	0.89(0.76,1.04)	0.137	0.90(0.76,1.05)	0.187
	rs243021	2	BCL11A	A/G	1.08(0.92,1.27)	0.362	1.07(0.89,1.27)	0.486	0.94(0.80,1.10)	0.436	0.94(0.80,1.11)	0.483
	rs2943641	2	IRS1	C/T	1.20(0.89,1.62)	0.226	1.26(0.92,1.73)	0.157	1.00(0.75,1.32)	0.985	0.98(0.74,1.31)	0.911
	rs3923113	2	GRB14	A/C	1.03(0.83,1.30)	0.770	1.02(0.80,1.30)	0.852	1.16(0.92,1.46)	0.210	1.15(0.91,1.45)	0.247
	rs7593730	2	RBMS1, ITGB6	C/T	1.08(0.87,1.33)	0.509	1.08(0.86,1.35)	0.532	0.91(0.74,1.12)	0.375	0.91(0.74,1.13)	0.399
	rs780094	2	GCKR	C/T	1.09(0.93,1.28)	0.284	1.10(0.93,1.30)	0.293	1.00(0.85,1.17)	0.964	0.97(0.83,1.14)	0.726
	rs1470579	3	IGF2BP2	C/A	0.98(0.82,1.18)	0.854	1.04(0.86,1.25)	0.716	1.04(0.87,1.24)	0.646	1.04(0.86,1.24)	0.708
	rs16861329	3	ST6GAL1	T/C	0.84(0.70,1.02)	0.081	0.89(0.72,1.09)	0.246	0.86(0.71,1.04)	0.118	0.85(0.70,1.03)	0.103
	rs4607103	3	ADAMTS9	C/T	1.13(0.97,1.33)	0.123	1.15(0.97,1.37)	0.099	0.87(0.74,1.01)	0.067	0.87(0.74,1.02)	0.083
	rs4858889	3	SCAP	A/G	1.01(0.81,1.26)	0.908	1.07(0.84,1.35)	0.587	0.93(0.75,1.16)	0.539	0.90(0.72,1.13)	0.358
	rs7612463	3	UBE2E2	C/A	0.89(0.73,1.07)	0.205	0.86(0.70,1.05)	0.142	1.04(0.86,1.25)	0.723	1.06(0.87,1.29)	0.561
	rs831571	3	PSMD6	C/T	1.10(0.93,1.29)	0.256	1.11(0.93,1.32)	0.262	1.02(0.87,1.20)	0.775	1.01(0.85,1.19)	0.956
	rs6815464	4	MAEA	C/G	0.85(0.73,1.00)	0.053	0.84(0.71,1.00)	0.044	1.12(0.96,1.32)	0.153	1.15(0.98,1.35)	0.098
	rs459193	5	ANKRD55	G/A	1.02(0.88,1.19)	0.780	1.07(0.91,1.26)	0.434	1.02(0.87,1.18)	0.854	1.01(0.87,1.19)	0.865
	rs10946398	6	CDKAL1	C/A	0.96(0.81,1.13)	0.592	0.93(0.79,1.11)	0.430	1.01(0.86,1.18)	0.921	1.03(0.88,1.22)	0.707
	rs1535500	6	KCNK16	T/G	0.93(0.80,1.09)	0.390	0.91(0.77,1.07)	0.239	1.06(0.91,1.23)	0.492	1.06(0.90,1.24)	0.486
	rs9470794	6	ZFAND3	C/T	0.97(0.82,1.14)	0.691	1.00(0.84,1.19)	0.973	1.10(0.93,1.29)	0.256	1.08(0.92,1.28)	0.343
	rs2191349	7	DGKB, TMEM195	T/G	1.07(0.91,1.25)	0.419	1.04(0.88,1.23)	0.648	0.93(0.79,1.08)	0.329	0.91(0.77,1.06)	0.225
	rs4607517	7	GCK	A/G	1.09(0.91,1.31)	0.354	1.06(0.87,1.29)	0.550	0.96(0.79,1.15)	0.629	0.97(0.81,1.18)	0.783
	rs864745	7	JAZF1	T/C	1.03(0.85,1.24)	0.783	1.01(0.82,1.23)	0.951	0.92(0.76,1.11)	0.367	0.95(0.79,1.15)	0.600
	rs972283	7	KLF14	G/A	0.92(0.77,1.09)	0.332	0.93(0.77,1.12)	0.449	0.99(0.84,1.18)	0.928	1.00(0.84,1.19)	0.969
	rs13266634	8	SLC30A8	C/T	1.07(0.91,1.25)	0.424	1.07(0.90,1.28)	0.416	1.11(0.95,1.30)	0.196	1.12(0.95,1.32)	0.164
	rs516946	8	ANK1	C/T	1.15(0.90,1.48)	0.271	1.27(0.97,1.65)	0.083	1.42(1.10,1.83)	0.008	1.39(1.07,1.81)	0.014
	rs896854	8	TP53INP1	T/C	0.95(0.81,1.12)	0.560	0.96(0.81,1.15)	0.683	0.99(0.84,1.16)	0.889	0.96(0.81,1.14)	0.639
	rs10811661	9	CDKN2A, CDKN2B	T/C	1.06(0.90,1.25)	0.500	1.10(0.93,1.31)	0.268	1.14(0.96,1.34)	0.131	1.15(0.97,1.36)	0.110
	rs17584499	9	PTPRD	T/C	1.10(0.86,1.41)	0.458	1.12(0.85,1.46)	0.427	0.90(0.70,1.17)	0.429	0.89(0.69,1.16)	0.404
	rs2796441	9	TLE1	G/A	0.91(0.78,1.06)	0.231	0.93(0.79,1.10)	0.387	1.03(0.88,1.21)	0.691	1.03(0.88,1.21)	0.680
	rs7041847	9	GLIS3	A/G	0.95(0.81,1.11)	0.526	0.98(0.83,1.16)	0.812	0.96(0.82,1.13)	0.627	0.93(0.80,1.10)	0.400
	rs10886471	10	GRK5	C/T	1.03(0.85,1.25)	0.748	1.07(0.87,1.31)	0.538	1.17(0.96,1.42)	0.118	1.20(0.99,1.47)	0.068
	rs10906115	10	CDC123, CAMK1D	A/G	0.91(0.77,1.07)	0.262	0.97(0.81,1.16)	0.747	1.13(0.96,1.33)	0.139	1.10(0.93,1.30)	0.260
	rs11257655	10	CDC123	T/C	0.90(0.77,1.06)	0.194	0.95(0.80,1.13)	0.538	1.12(0.96,1.31)	0.159	1.08(0.92,1.27)	0.366
	rs5015480	10	HHEX	C/T	1.18(0.96,1.44)	0.111	1.24(1.00,1.53)	0.054	0.90(0.73,1.10)	0.305	0.86(0.69,1.06)	0.161
	rs10830963	11	MTNR1B	G/C	0.99(0.84,1.16)	0.876	1.06(0.89,1.25)	0.524	1.09(0.93,1.27)	0.304	1.08(0.92,1.27)	0.371
	rs1552224	11	CENTD2	A/C	0.75(0.58,0.98)	0.037	0.77(0.58,1.01)	0.063	1.39(1.03,1.86)	0.030	1.47(1.09,1.99)	0.013
	rs2237892	11	KCNQ1	C/T	0.97(0.82,1.15)	0.719	0.97(0.81,1.16)	0.721	1.10(0.92,1.30)	0.300	1.12(0.94,1.34)	0.207
	rs5215	11	KCNJ11	C/T	0.93(0.79,1.09)	0.362	0.90(0.76,1.07)	0.225	1.13(0.97,1.33)	0.123	1.15(0.97,1.35)	0.100
	rs10842994	12	KLHDC5	C/T	1.09(0.89,1.34)	0.386	1.12(0.91,1.39)	0.293	0.95(0.78,1.15)	0.577	0.95(0.77,1.16)	0.606
	rs7961581	12	TSPAN8, LGR5	C/T	0.86(0.71,1.05)	0.137	0.88(0.71,1.08)	0.208	1.25(1.04,1.50)	0.020	1.25(1.03,1.51)	0.022
	rs11634397	15	ZFAND6	G/A	1.04(0.81,1.34)	0.742	1.10(0.84,1.44)	0.488	0.89(0.68,1.15)	0.365	0.87(0.67,1.13)	0.287
	rs2028299	15	AP3S2	C/A	1.07(0.89,1.29)	0.472	1.09(0.90,1.32)	0.391	1.19(0.99,1.42)	0.065	1.17(0.97,1.41)	0.100
	rs7172432	15	C2CD4A, C2CD4B	A/G	1.16(0.98,1.36)	0.082	1.13(0.95,1.35)	0.156	1.15(0.98,1.35)	0.092	1.12(0.95,1.33)	0.166
	rs7178572	15	HMG20A	G/A	1.06(0.90,1.25)	0.480	1.09(0.92,1.30)	0.331	1.11(0.94,1.30)	0.219	1.09(0.92,1.28)	0.337
	rs7403531	15	RASGRP1	T/C	1.32(1.12,1.56)	0.001	1.35(1.13,1.62)	0.001	0.95(0.81,1.13)	0.565	0.93(0.78,1.10)	0.373
	rs1558902	16	FTO	A/T	1.09(0.87,1.38)	0.454	1.05(0.82,1.34)	0.72	1.08(0.86,1.36)	0.507	1.08(0.86,1.37)	0.504
	rs7202877	16	BCAR1	T/G	0.93(0.76,1.13)	0.443	0.93(0.75,1.15)	0.505	0.88(0.72,1.07)	0.184	0.89(0.73,1.08)	0.231
	rs8050136	16	FTO	A/C	1.08(0.85,1.36)	0.535	1.03(0.80,1.33)	0.805	1.09(0.86,1.37)	0.484	1.08(0.85,1.37)	0.525
	rs4430796	17	HNF1B	G/A	0.99(0.83,1.17)	0.884	0.96(0.80,1.16)	0.676	0.98(0.83,1.16)	0.824	0.98(0.82,1.17)	0.825
	rs12454712	18	BCL2	T/C	1.02(0.87,1.19)	0.852	1.00(0.85,1.18)	0.966	1.00(0.86,1.17)	0.997	0.98(0.84,1.15)	0.815
	rs12970134	18	MC4R	G/A	0.93(0.76,1.15)	0.510	0.95(0.76,1.19)	0.656	1.10(0.89,1.36)	0.365	1.11(0.89,1.37)	0.355
	rs8090011	18	LAMA1	G/C	0.92(0.77,1.09)	0.339	0.94(0.78,1.13)	0.476	1.10(0.93,1.32)	0.276	1.11(0.93,1.33)	0.249
	rs10401969	19	CILP2	C/T	1.03(0.79,1.36)	0.826	1.00(0.74,1.33)	0.977	0.99(0.75,1.30)	0.937	0.98(0.74,1.30)	0.907
	rs3794991	19	GATAD2A	T/C	1.14(0.84,1.56)	0.392	1.12(0.80,1.56)	0.510	0.87(0.63,1.20)	0.392	0.86(0.61,1.20)	0.362
**Birth weight**	rs724577	4	LCORL	C/A	0.98(0.83,1.15)	0.764	1.04(0.88,1.25)	0.632	1.06(0.90,1.25)	0.486	1.03(0.87,1.21)	0.758
	rs4432842	5	5q11.2	T/C	1.17(1.00,1.37)	0.050	1.23(1.04,1.45)	0.017	1.03(0.88,1.21)	0.701	1.04(0.89,1.22)	0.609
	rs6931514	6	CDKAL1	G/A	1.00(0.85,1.19)	0.976	1.01(0.85,1.21)	0.882	0.93(0.79,1.10)	0.390	0.95(0.80,1.13)	0.562
	rs1042725	12	HMGA2	T/C	1.00(0.83,1.22)	0.989	0.98(0.8,1.21)	0.864	0.98(0.81,1.19)	0.840	0.96(0.79,1.17)	0.716
**BMI**	rs2568958	1	1p31	A/G	1.08(0.82,1.43)	0.582	1.10(0.81,1.49)	0.545	0.86(0.66,1.13)	0.276	0.85(0.64,1.11)	0.233
	rs574367	1	SEC16B	G/T	1.25(1.02,1.53)	0.033	1.34(1.07,1.67)	0.010	1.01(0.83,1.23)	0.925	1.01(0.83,1.24)	0.893
	rs7561317	2	TMEM18	G/A	0.95(0.73,1.24)	0.725	0.98(0.74,1.3)	0.898	1.07(0.81,1.39)	0.645	1.11(0.84,1.46)	0.468
	rs16892496	8	TRHR	C/A	0.98(0.84,1.14)	0.783	1.00(0.85,1.18)	0.968	0.91(0.78,1.06)	0.224	0.90(0.77,1.05)	0.179
	rs7832552	8	TRHR	T/C	0.96(0.82,1.12)	0.585	0.98(0.83,1.15)	0.782	0.90(0.78,1.05)	0.193	0.89(0.76,1.04)	0.142
	rs11030104	11	BDNF, BDNF-AS	A/G	1.18(1.01,1.39)	0.041	1.12(0.95,1.33)	0.184	0.96(0.82,1.12)	0.582	0.95(0.81,1.11)	0.515
	rs6265	11	BDNF, BDNF-AS	C/T	1.14(0.97,1.34)	0.108	1.10(0.93,1.31)	0.263	0.95(0.81,1.11)	0.524	0.94(0.80,1.11)	0.467
	rs9939609	16	FTO	A/T	1.02(0.81,1.30)	0.846	0.97(0.75,1.25)	0.803	1.06(0.84,1.35)	0.612	1.06(0.83,1.35)	0.641

Model 1 was logistic regression model to analyze the association between individual SNP and quantitative traits without adjusting covariates. Model 2 was logistic regression model to analyses the association between individual SNP and quantitative traits with adjusting continuous covariates, BMI, waist circumference, and sedentary time, and categorical covariates, gender, economical status, education levels, exercise, whole cereal and beans intake levels, livestock and poultry intake levels, drinking, and smoking.

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
