# Peer review of "A Replication Study Identified Seven SNPs Associated with Quantitative Traits of Type 2 Diabetes among Chinese Population in A Cross-Sectional Study"

_ijerph, 2020, doi:10.3390/ijerph17072439_

Round 1

Reviewer 1 Report

The manusript by Yuan et al on diabetes associated SNPs describes links between certain gene polymorphisms and the risk of type 2 diabetes (T2D) and T2D related traits. While the topic of the manuscript is interesting, the presentation of the results could be improved.
The manuscript would benefit if its results were presented in a more visual form i.e - graphs and diagrams instead of simple tables. The association between any given SNP and an increseased risk of T2D-related traits should be make more clear and presented in a form of a graph. The paper also needs English language revision. Many sentences are missing verbs or need to be reformated (some obvious examples are highlighted and commented in the attachment)

Author Response

We are extremely grateful for your positive and constructive feedback. Please see the attachment for our response to your each comment.

Reviewer 2 Report

The authors present the results of a candidate gene cohort study on glycometabolic traits in subjects of asian ethnicity.

The abstract should include not only the sample size selected for analysis (2216), but also the considerably smaller actual sample size suitable for statistical analysis. (1677)

Introduction and methodology are sound, results are presented in an appropriate manner, the discussion is adequately adressing the major results.

In Lines 99 ff. the authors link 22/63 SNP to beta cell failure, 28/63 to insulin resistance, but don't classify the remaining 13 SNPs. Please clarify.

In Lines 113 ff. intake of specific food items is classified as insufficient or (very) sufficient. As there is no approved cut-off for the mandatory or recommended intake of cereals, beans or animal-based food, the labelling should be changed to low, medium, high.

Please indicate how you corrected for multiple comparison.

Table 1 shows a binary classification for HOMA-IR and HOMA-B. Please state the used cut-offs.

In the limitations section, the quite small percentage of T2DM patients, the skewed sex ratio, and the narrow age range (usually too young to get T2DM) should be mentioned as well.

The conclusions should not be limited to 4 out of 7 significant SNPs.

Language, grammar, punctuation, and fonts should be revised thoroughly.

Significant associations in Table 2 should be highlighted in bold, with an asterisk or otherwise.

Author Response

(The authors gave the same response as above.)

Round 2

Reviewer 2 Report

Thanks for this thorough revision. Most of the points from the first review have been addressed sufficiently.

Few points are remaining unanswered:

1) The analysis is lacking a correction for multiple comparison (Bonferroni, Benjamini-Hochberg...).

2) Classification of HOMA-IR and HOMA-beta by percentiles is not useful, as the cohort contains mostly healthy persons, and there are approved and widely accepted cut-offs for HOMA-IR and HOMA-beta in the literature. Those cut-offs should be used.

Author Response

We are extremely grateful for your positive and constructive feedback again. Here bellows are our response to your each comment

Comment 1: The analysis is lacking a correction for multiple comparison (Bonferroni, Benjamini-Hochberg...).

Response: We used logistic regression to test association of individual SNP with T2D-related quantitative traits, rather than all 64 SNPs entering the model together. Therefore, the correction for multiple comparison was not necessary.

Comment 2: Classification of HOMA-IR and HOMA-beta by percentiles is not useful, as the cohort contains mostly healthy persons, and there are approved and widely accepted cut-offs for HOMA-IR and HOMA-beta in the literature. Those cut-offs should be used.

Response: The previous studies had indicated that the cut-off values of HOMA-IR varied by different races,ages, genders, diseases, etc. and few study defined impaired insulin release according to HOMA-β. While in China,no widely accepted cut-offs for HOMA-IR and HOMA-β is available. Therefore, our study adopts the definition of insulin resistance and impaired insulin release by Liu Rong et al research.

Round 3

Reviewer 2 Report

All comments have been addressed sufficiently. I recommend to accept the manuscript in the present form.